# speedingCARs: accelerating the engineering of CAR T cells by signaling domain shuffling and single-cell sequencing

Rocío Castellanos-Rueda [1,2,4], Raphaël B. Di Roberto [1,4], Florian Bieberich [1,2], Fabrice S. Schlatter [1], Darya Palianina [3], Oanh T. P. Nguyen [1], Edo Kapetanovic [1], Heinz Läubli [3], Andreas Hierlemann [1], Nina Khanna [3] & Sai T. Reddy [1] ✉

Chimeric antigen receptors (CARs) consist of an antigen-binding region fused to intracellular signaling domains, enabling customized T cell responses against targets. Despite their major role in T cell activation, effector function and persistence, only a small set of immune signaling domains have been explored. Here we present speedingCARs, an integrated method for engineering CAR T cells via signaling domain shuffling and pooled functional screening. Leveraging the inherent modularity of natural signaling domains, we generate a library of 180 unique CAR variants genomically integrated into primary human T cells by CRISPR-Cas9. In vitro tumor cell co-culture, followed by single-cell RNA sequencing (scRNA-seq) and single-cell CAR sequencing (scCAR-seq), enables high-throughput screening for identifying several variants with tumor killing properties and T cell phenotypes markedly different from standard CARs. Mapping of the CAR scRNA-seq data onto that of tumor infiltrating lymphocytes further helps guide the selection of variants. These results thus help expand the CAR signaling domain combination space, and supports speedingCARs as a tool for the engineering of CARs for potential therapeutic development.

Cellular immunotherapies against cancer have made substantial progress in recent years, with six FDA-approved chimeric antigen receptor (CAR) T cell treatments against hematological malignancies. These treatments rely on synthetic protein receptors that have been engineered for precise molecular recognition of a cell surface antigen (e.g., CD19 on the surface of B cell lymphomas). The infusion of autologous CAR T cells results in a cytotoxic response against tumor cells and, as in a classical immune reaction, this treatment can potentially result in persistent immunity, with CAR T cells recently observed in patients several years post-treatment[1]. However, success in this field has been difficult to replicate outside of hematological B cell malignancies. For example, solid tumor cancers, such as in the breast or lung, are more resistant to CAR T cell-mediated killing and have struggled to make progress clinically. Instances of relapse, sometimes through antigen escape[2] have also highlighted the limitations of CAR T cell persistence and of the monoclonality of the infusion product. Furthermore, strong adverse events, such as cytokine release syndrome (CRS) and transient neurotoxicity[3] are frequently associated with treatment and thus represent considerable safety concerns. Together, these pitfalls of CAR T cell therapies form substantial obstacles to their broader use against a wider range of cancer types.

[1]Department of Biosystems Science and Engineering, ETH Zürich, 4058 Basel, Switzerland. [2]Life Science Zurich Graduate School, ETH Zürich, University of Zurich, 8057 Zürich, Switzerland. [3]Department of Biomedicine, University of Basel, 4031 Basel, Switzerland. [4]These authors contributed equally: Rocío Castellanos-Rueda, Raphaël B. Di Roberto. ✉e-mail: sai.reddy@ethz.ch

In order to enhance CAR T cell responses against tumors, recent work has leveraged immunological mechanisms to counter the immunosuppressive microenvironment. For example, co-administration with immune-checkpoint blockade inhibitors (ICB; e.g., monoclonal antibodies targeting PD-1/PD-L1) can activate tumor-infiltrating T cells[4]; or alternatively, armoring CAR T cells with additional genetic modifications to make them overexpress certain cytokines (e.g., IL-12, IL-23, etc.) can potentially sensitize tumors to cell-mediated cytotoxicity[5]. Furthermore, synthetic biology approaches that incorporate controllable domains[6–8] are being utilized to design CARs responsive to drugs or antigenic cues in order to tune the strength, duration and specificity of the inflammatory response. While promising, all of these strategies require additional drug compounds or genetic modifications, introducing further complexity to a therapeutic regimen that is already laborious and sensitive.

Recent advances in molecular immune profiling, such as single-cell sequencing and transcriptome analysis, are contributing important quantitative insights on CAR T cell-mediated responses and patient outcomes[9]. By going beyond methods for standard cell phenotyping (e.g., detection of surface markers by flow cytometry), transcriptional phenotyping can uncover gene expression patterns and metabolic pathways, or when used to track therapy progression and outcomes, can help to identify predictors of therapeutic efficacy. A key observation from such studies is that the molecular components and design of a CAR have a major influence on the features of the transcriptional response[10]. A conventional CAR is rationally designed based on the fusion of modular gene elements: an extra-cellular antigen-recognition domain (frequently an antibody single chain variable fragment (scFv)), structural elements (a hinge, a transmembrane helix and peptide linkers), as well as one or more intracellular signal-transducing domains. The signaling domains are essential for linking the antigen-cell binding event to a cascade of intracellular molecular events, culminating in the expression of pro-inflammatory and cytotoxic genes. As would be expected, the signaling domains used can impact the type of response induced. However, almost all CARs in clinical trials to date combine the signaling domain CD3ζ of the T cell receptor (TCR) complex and the signaling domains of the co-receptors CD28 and 4-1BB[11]. Only a small panel of alternative signaling domains has been individually investigated clinically, and even fewer in parallel[10,12,13]. This lack of diversity is partly due to the low-throughput aspect of rational CAR design as well as the laboriousness of performing in vitro functional assays. Recently, two studies from Gordon et al.[14] and Goodman et al.[15] developed approaches to generate pooled CAR signaling domain libraries, which were screened by combining fluorescence-activated cell sorting (FACS) using standard T cell markers and amplicon sequencing to identify novel functional variants[14,15].

Here, we describe speedingCARs: single-cell sequencing of pooled engineered signaling domain libraries of CARs. Natural immune signaling domains are combinatorially shuffled to generate a highly diverse library of CAR variants. This library is expressed in primary human T cells by genome editing and co-cultured with tumor targets to screen variant's functional capabilities, comparing their induced single-cell transcriptional profiles. The resulting high-quality and high-resolution data is mapped onto the transcriptional landscape of tumour-infiltrating lymphocytes (TILs) derived from lung tumors of treatment-responsive patients. The combination of single-cell RNA sequencing (scRNA-seq), single-cell CAR sequencing (scCAR-seq), together with clinical data mapping serves to guide the selection of promising variants which are then functionally validated based on cytokine secretion, differentiation and cell-mediated cytotoxicity. Thus, speedingCARs can be used to rapidly expand the diversity of the CAR synthetic protein family, offering a range of flexible immune phenotypes to tackle immunotherapeutic challenges.

## Results

### Design, generation and expression of a CAR signaling domain library in primary human T cells

In order to generate a library of CAR variants that could induce diverse T cell transcriptional phenotypes, we designed a modular cloning and assembly strategy for shuffling intracellular signaling elements (Fig. 1a). In a standard CAR, the intracellular region is composed of a CD3ζ signaling domain on the C-terminal end and co-receptor domains CD28 and/or 4-1BB on the N-terminal end (proximal to the transmembrane domain). CD3ζ is notable among other immune signaling proteins by the presence of three immunoreceptor tyrosine activation motifs (ITAMs), which play an important role in triggering downstream signaling events in response to receptor clustering. Although co-receptor domains such as CD28 and 4-1BB do not have ITAMs, their presence mimics the co-stimulation that accompanies TCR engagement with an activated antigen-presenting cell (APC), resulting in a more durable response.

Although recent work has highlighted some degree of plasticity in the number and configuration of ITAMs[16], our library design retained a total sum of three ITAMs to maximize functionality. To ensure this, we segmented the CD3ζ gene between the first and second ITAM and retained the two-ITAM segment for the CAR library. We then selected intracellular domains from a variety of immune co-receptors for inclusion into two pools of gene segments: domains A and B (Fig. 1b). The pool of domain A was generated from 15 receptors that have been described as providing co-stimulation in immune cell-cell interactions, ranging from the major contributors CD28 and 4-1BB to potentially more minor receptors, such as CD30 or CD150 and CD84 of the signaling lymphocytic activation molecule (SLAM) family. We also included lesser-known or inhibitory receptors such as FCRL6, CD244 and LAG3 to investigate their synergistic effects[17]. The pool of domain B consisted of 12 genes that each possessed a single ITAM. This included receptors such as FcγRIIA, DAP12 and CD79B, which are expressed in diverse immune cells from both myeloid and lymphoid lineages, as well as T cell-specific genes involved in the TCR complex such as CD3G. In addition, we included three viral receptors, LMP2 (Epstein-Barr virus), K1 (Kaposi's sarcoma-associated herpesvirus) and GP (New York virus), to investigate whether their non-human origin could lead to unique functional response profiles. Lastly, we also used the first segment of CD3ζ, which would reconstitute the full intracellular domain upon assembly.

To shuffle and assemble the domain gene segments within the CAR genetic chassis, we used a Type IIS restriction cloning strategy[18] (Fig. 1c). In this method, customized restriction sites within a cloning cassette ensure that domains assemble in an orderly fashion: assembly of domain A on the N-terminal side and domain B on the C-terminal side results in the three ITAMs being proximal to each other. The two variable signaling domains are joined by a minimal linker. The remainder of the CAR chassis consisted of the following constant elements: (i) a secretion signal peptide from CD8α; (ii) an extracellular scFv based on the monoclonal antibody trastuzumab (scFv clone: 4D5); (iii) a CD28-derived hinge and transmembrane domain; (iv) the partial segment of CD3ζ bearing two-ITAMs. Through the scFv, all CAR variants in the library have binding specificity for the oncogenic human epidermal growth factor receptor 2 (HER2), a clinically relevant target for CAR T cell therapy due to its prominence in many cancers[19]. Following cloning in a plasmid vector, the diversity of the resulting signaling domain library was assessed by deep sequencing, showing that the library possessed 179/180 possible combinations, with balanced representation (Fig. 1d).

To express the library of CAR variants in a physiologically relevant context, we performed genome editing with CRISPR-Cas9 on primary human T cells from healthy donors. Homology-directed repair (HDR) was used to target the CAR library for genomic integration at the TCR alpha chain (TRAC) locus (Fig. 1e). The precise integration of CARs at

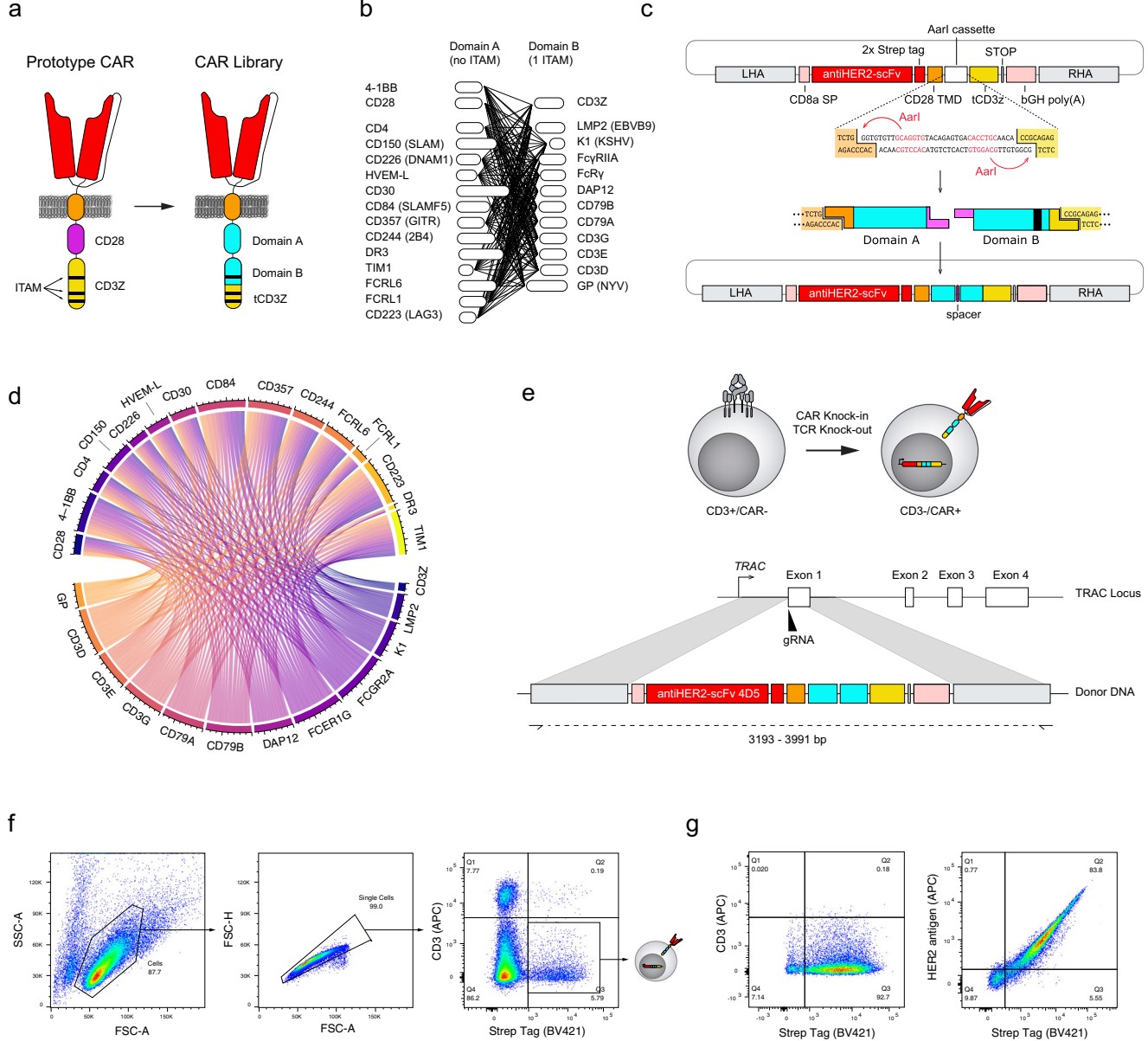

**Fig. 1 | Combining modular domain shuffling and genome editing to express a library of CAR variants in primary human T cells. a** Schematic representation of CAR architecture for the shuffling of signaling domains. The CAR variant library is derived from an initial second-generation CAR featuring the intracellular signaling domains of CD28 and CD3ζ; the scFv (4D5) is based on the variable domains of the clinical antibody trastuzumab (specificity to the antigen HER2). The entire CD28 signaling domain and a segment of the CD3ζ domain are exchanged with signaling domains from two pools: Domain A and B, which possess either zero or one ITAM, respectively. A truncated CD3ζ (tCD3Z) possessing two ITAMs is retained. **b** Combinatorial shuffling of Domain A and Domain B intracellular signaling domains yields a library of 180 possible CAR variants. **c** Schematic representation of the cloning strategy for domain shuffling. A backbone vector was designed to encode a CAR chassis composed of the following conserved elements: the CD8α secretion peptide, the scFv 4D5, two Strep tags separated by G₄S linkers,

the CD28 hinge and transmembrane domains and a partial region of CD3ζ. A cloning cassette between the transmembrane domain and CD3ζ domain has outward-facing recognition sequences of the Type IIS restriction enzyme *AarI*. A restriction digest yields unique overhangs that are compatible with the ligation of one domain from pool A and one domain from pool B, in that order (5' to 3'). The construct is completed by a polyadenylation signal and flanked by homology arms for the targeted genomic integration of the transgene in the human *TRAC* locus. **d** Long-read deep sequencing was performed following cloning and assembly of the signaling domain shuffling library. Circos plot shows that 179/180 possible combinations are present in balanced proportions (Source data are provided as a Source Data file.). **e** Schematic of the strategy for targeted genomic integration of a CAR library into the *TRAC* locus of primary human T cells by CRISPR-Cas9 HDR. **f, g** Flow cytometry sequential gating of human primary T cells after transfection (**f**) and after selection for CAR expression (**g**).

the *TRAC* locus has previously been shown to enhance tumor cell killing while conferring notable advantages over viral gene delivery (e.g., retrovirus, lentivirus): it ensures most T cells are monoclonal (only a single and unique CAR); it leads to transgene expression that is more consistent across cells and physiologically regulated, and it knocks out the endogenous TCR to avoid confounding effects[20].

Following genome editing, CAR T cells were isolated from non-edited cells by FACS based on surface expression of a Strep tag and deletion of the TCR (Fig. 1f). The post-sort purity of the CAR T cell population and its ability to bind to soluble HER2 antigen was confirmed by flow cytometry (Fig. 1g), while genotyping validated targeted integration into the *TRAC* locus (Supplementary Fig. 1).

## Functional and pooled screening of CAR T cell library by single-cell sequencing

In order to achieve functional and pooled screening of the CAR T cell library, we performed co-cultures with tumor cells followed by scRNA-seq. This enabled us to trigger the activation of the CAR signaling domain variants and to profile the associated transcriptional phenotypes and key gene expression signatures. This approach provides sufficient coverage of a CAR library of 180 unique variants by taking advantage of the current capacity of $10^3$–$10^4$ cells for scRNA-seq (e.g., 10X Genomics, as used here). In addition, we designed scCAR-seq, a strategy to de-multiplex the CAR library from the resulting scRNA-seq data. For this, CAR single-cell barcoded transcripts found in the scRNA-seq cDNA product are selectively amplified and long-read deep sequencing (Pacbio) is used to capture the barcode identifiers that link the CAR signaling domain variant to a given cell (Supplementary Fig. 2).

Primary human T cells expressing the CAR library were co-cultured with the HER2-expressing breast cancer cell line SKBR3. To serve as a benchmark in subsequent analyses, we also spiked-in T cells bearing clinically used standard CARs with the CD28-CD3ζ and 4-1BB-CD3ζ domain combinations (abbreviated to 28z and BBz) into the pooled co-cultures. Negative controls consisted of co-cultured TCR-negative T cells (generated by Cas9-induced non-homologous end joining (NHEJ) in the *TRAC* locus; TCR-) and unstimulated (no tumor co-culture) 28z CAR T cells. Following co-culture (36 h), CAR T cells were isolated by FACS (Supplementary Fig. 3) and processed for scRNA-seq through the 10X Genomics pipeline. RNA transcripts were barcoded and amplified as cDNA to generate a gene expression (GEX) library suitable for deep sequencing. In parallel, scCAR-seq was performed and the resulting amplified transcripts were used for long-read deep sequencing.

## Identification of CAR-specific induced transcriptional phenotypes

Following pre-processing and quality filtering, a total of 19,321 stimulated CAR T cells across three donors were sequenced by scRNA-seq. Among these, 55% could be annotated with a single specific CAR variant, while 4% had to be discarded due to the identification of more than one different CAR construct in the same cell. This resulted in a total of 10,692 annotated cellular transcriptomes (averaging ~$2.5 \times 10^4$ reads/cell) covering 156 unique CAR variants from our library. Although we selected a relatively short co-culture time of 36 h in order to minimize more proliferative variants from becoming over-represented, we saw evidence of clonal variant expansion (Supplementary Fig. 4a). In order to do a clustering and gene expression analysis encompassing both over- and under-represented variants, we randomly subsampled up to 250 cells per variant and 500 cells for each negative control sample. To confirm that this did not significantly affect clustering, 50 independent iterations of the subsampling and clustering algorithms were performed and the adjusted Rand index (ARI) was computed for each pair of partitions. This measure of data clustering similarity was consistently high, confirming that no specific subsample would likely change our results (Supplementary Fig. 4b–d). The reduced dataset was used to do cell clustering and CAR variant enrichment analysis thereafter.

To determine whether CAR variants could trigger different functional T cell states upon encounter with tumor cells, we assessed the enrichment of each variant across T cell phenotypes. Unsupervised clustering identified thirteen clusters which were annotated based on CD4 and CD8 expression, predicted cell cycle phase and the differential gene expression of T cell marker genes (Fig. 2a, b, and Supplementary Fig. 5a–c); CD8+ cells were divided into a memory cluster characterized by having a low number of cycling cells and high expression of CCR7, IL7R, CD27, CD7, LEF1 and TCF7 genes; a proliferative BATF3 + cluster with low expression of cytotoxicity and T cell

activation markers and enrichment in stress response and HLA genes; and 4 clusters presenting effector-like features: Effector A (KLRD, GNLY, and PRF), Effector B (NKG7,GZMK,GZMH, and EOMES), Cytotoxic (GZMB, IFNG, TNFRSF9, CCL3, CCL4, CSF2, XCL1, and CRTAM) and Terminal effector (TIGIT, ENTPD1, NPW, SOX4, LAIR2). CD4+ clusters were annotated as resting and cycling memory (expressing STAT1 and the previously described memory markers), activated memory (CCR7, IL2RA, LIF, ZBED2), KLF2 + resting, activated (IL26, IL17A, IL4I1, TNFRSF4, and TNFRSF18) and effector (DUSP4, ID3, TIGIT, and NPW). Finally, we observed a metabolically active mix of CD4/CD8 cells.

After defining the main cell clusters, the enrichment of CAR variants across clusters was evaluated to assess their effect on T cell activation. With few exceptions, most CAR variants were found to be distributed across multiple clusters (Fig. 2c and Supplementary Fig. 6). This was also the case for positive (28z and BBz) and negative control groups (TCR-negative and unstimulated 28z). However, five clusters were strongly enriched in the former and depleted in the latter (Fig. 2d, e). The low presence of TCR-negative and unstimulated 28z cells in these clusters suggests that they are associated with CAR induced stimulation (CAR Induced Clusters; CICs). The other eight clusters could result from either CAR stimulation and/or residual signaling received during standard in vitro T cell manipulation (CD3Z/CD28 bead activation and IL-2 treatment). Pathway enrichment analysis of differentially-expressed genes between CICs and non-CICs support this: non-CICs were enriched in pathways associated with CD3Z/CD28 stimulation, such as TCR binding and downstream signaling, antigen presentation, MHC protein interaction and CD28 costimulation, as opposed to CICs, which upregulated genes enriched in a broader set of pathways including cytokine, chemokine and receptor signaling (Supplementary Fig. 7). The enrichment of CAR variants in CICs was therefore used as a measure of CAR activity and to further investigate their profiles (Fig. 2f). Several CAR candidates appeared enriched in the CD8 Terminal Effector cluster (e.g. CD30-CD79B and CD4-K1). The CD8 Cytotoxic cluster was preferentially enriched by 28z compared to BBz cells, consistent with the enhanced cytotoxic potency of the former, and accounted for a large fraction of cells in variants such as CD28-FCGR2A and FCRL6-CD3G. Regarding CD4+ clusters, CD4 Activated Memory was more enriched in 28z and BBz cells compared to CD4 Cycling Memory, which seemed to be more expanded in variants such as 4-1BB-FCER1G and TIM1-CD3D. The CD4 effector cluster was exclusively enriched in 4-1BB-CD79A. In addition, a few CAR variants showed close to no enrichment in any of the CICs (e.g. TIM1-CD79B, CD357-CD79B and CD30-CD3D). The similarity of these cells to CAR unstimulated cells might indicate reduced potency of these CARs. Overall, the complex distribution of CAR variants across clusters (Supplementary Fig. 8) and in particular within CICs illustrates the diverse range of phenotypes that can be triggered by alternative CAR architectures.

## Examining CAR variant-specific transcriptional signatures

The classification of cells into clusters can identify general patterns in the data. However, computational algorithms delineate clusters in an unsupervised manner, which may not capture intra-cluster differences found in such a heterogeneous population of T cells. For this reason, we next examined the gene expression signature of each CAR variant individually using the full dataset of sequenced cells. A pseudo-bulking strategy was used to compare CAR variants with the standard 28z and BBz CARs and TCR-negative cells. Principal component analysis (PCA) revealed that different signaling domain combinations induce different T cell signatures; individual variants were found to be spread across the T cell phenotypic space (Fig. 3a). As expected, PC1 and PC2 clearly separate positive controls 28Z and BBz from the negative control TCR-negative. In addition, we observed a trend in the distribution of variants across the two main principal components and

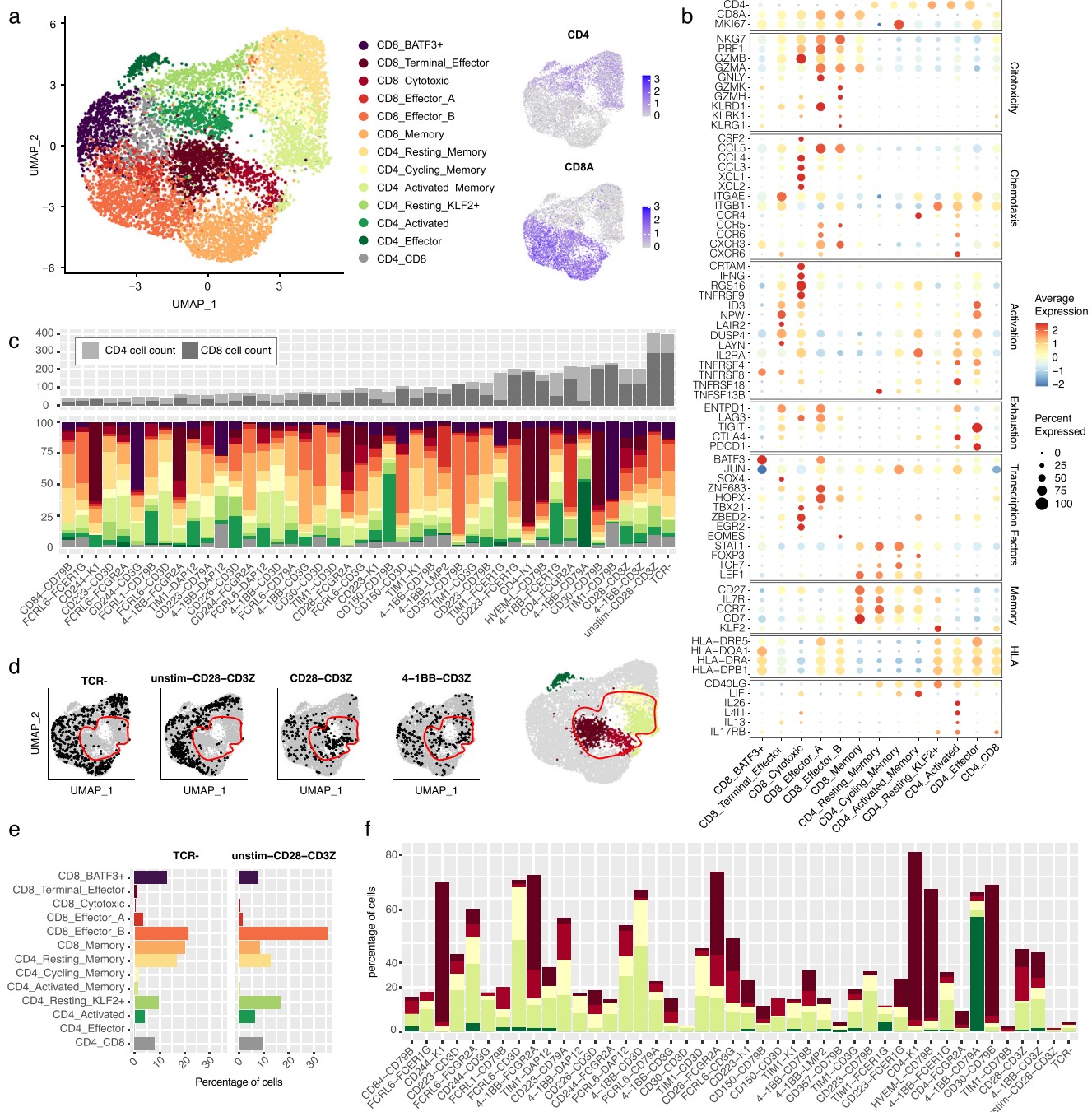

**Fig. 2 | Unsupervised cell clustering classifies CAR variants based on distinct transcriptional phenotypes. a** UMAP embedding and unsupervised cell clustering based on scRNA-seq data of the pooled CAR T cell library following functional activation (co-culture with cognate antigen-expressing SKBR3 tumor cells). Shown are data from 7,244 variant-assigned CAR T cells with random subsampling restricting it to a maximum of 250 cells per CAR construct. In addition, 500 cells for TCR-negative (TCR-) and unstim-CD28-CD3Z negative control samples were included. On the right, UMAP feature plots show the distribution of CD4- and CD8-expressing cells. **b** Dot plot showing the expression of a selection of T cell marker genes across clusters found in **a**. Color and size indicate differences in expression levels between groups. **c** Cluster enrichment observed for the top 40 most represented CAR variants (variants with at least 50 cells assigned), benchmark controls

(CD28-CD3Z and 4-1BB-CD3Z) and negative controls (TCR- and unstim-CD28-CD3Z). Variants are ordered by a confidence score based on the number of available cells. The top panel displays the CD4 and CD8 counts for each group and the bottom panel the fraction of cells found in each of the clusters from **a**. **d** Distribution of the unstimulated controls and the benchmark controls within the UMAP embedding of **a**. On the right a UMAP embedding plot highlights the CICs. In all UMAP plots, a red line depicts the region where there was no enrichment of unstimulated control samples. **e** Bar plot showing the distribution of CAR unstimulated control cells between the different clusters. **f** Bar plot depicting the percentage of cells per CAR variant that belong to the 5 different CICs. Source data for panels **c**, **e**, **f** are provided as a Source Data file.

their enrichment in CICs. While PC1 drives the separation of samples based on memory vs effector features, PC2 separates samples based on their enrichment in CICs. It is reasonable to infer that proximity to 28z and BBz controls could be an indicator of CAR activity, while

candidates that remain close to TCR-negative probably have limited antitumor potential.

In a similar way, comparing CD8+ pseudo-bulked samples of a selection of variants against the clinically standard CARs and

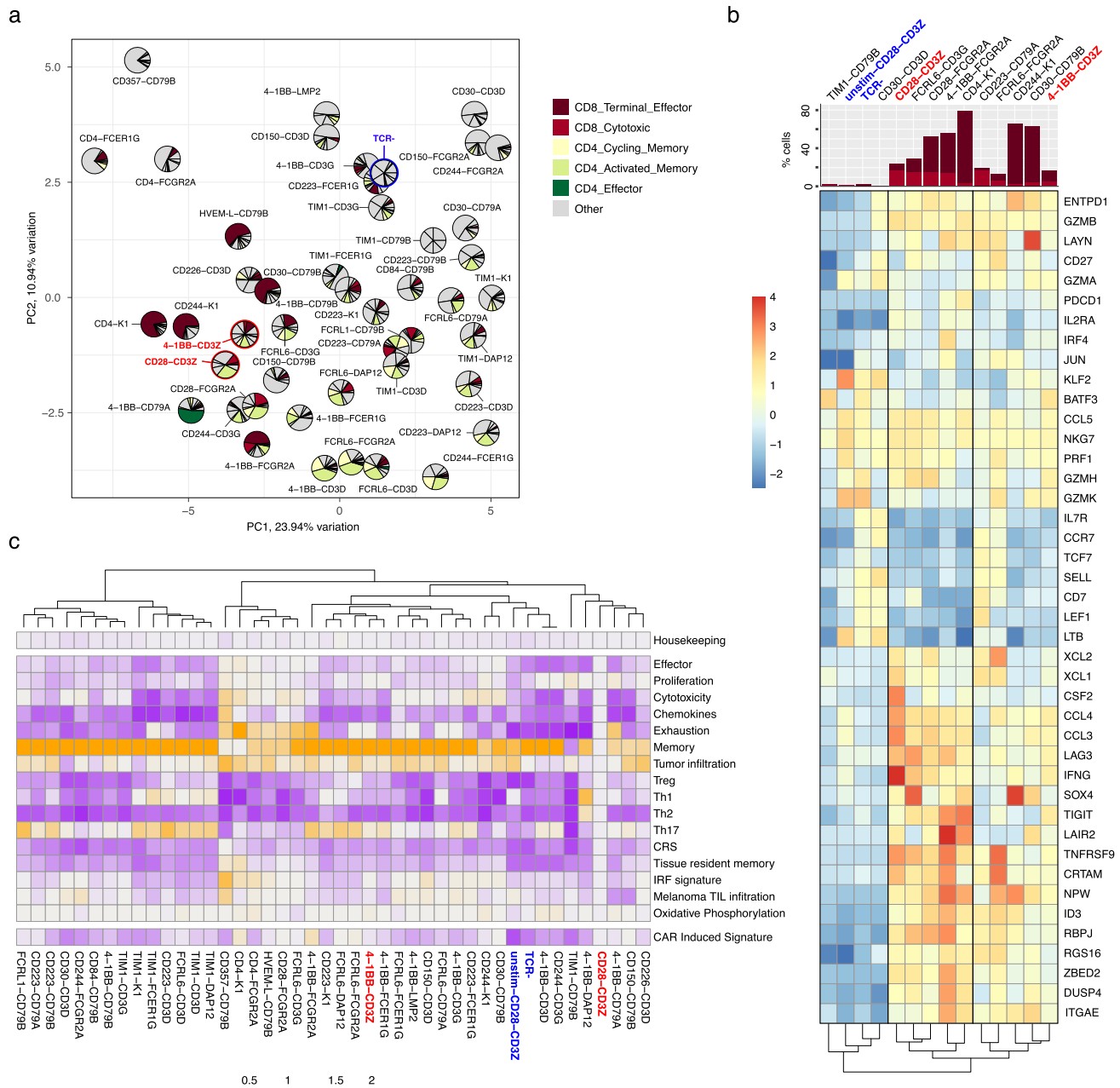

**Fig. 3 | Specific gene transcriptional signatures are used to assess CAR T cell functionality. a** Principal component analysis (PCA) of pseudo-bulked scRNA-seq data from the top 40 most represented CAR variants studied in Fig. 2. Also included are cells expressing 28z (CD28-CD3Z) and BBz (4-1BB-CD3Z) colored in red and CAR negative T cells (TCR-) colored in blue. To avoid batch effect variation, only data from Donor 3 is used. Overlayed over each data point, pie charts represent the enrichment of cells in CICs from Fig. 2f. **b** Expression levels of a set of 42 T cell marker genes across CD8+ pseudo-bulked scRNA-seq samples of a small panel of 10 different CAR T cell variants, 28z and BBz benchmark CAR T cells, TCR- T cells and unstimulated 28z CAR T cells. CAR variants were selected based on their distinct enrichment in CD8 CICs. Marker genes describe phenotypes such as memory, activation, cytotoxicity and exhaustion. Color indicates normalized gene expression deviation from average. Genes and samples are ordered by hierarchical clustering. On the top, a bar plot indicates the percentage of cells identified to be part of the two main CD8+ CICs. **c** Heatmap showing the difference in gene-set scores (score based on the simultaneous expression of different gene sets computed for each single cell) between the 40 most represented CAR variants, 28z and BBz CARs, unstimulated 28z CAR and CAR negative T cells, for 18 different gene sets. The mean score per variant is given as a fold change measurement when compared to 28z WT CAR.

negative control groups revealed differences across key genetic markers (Fig. 3b). Hierarchical clustering based on the gene expression levels of 42 T cell marker genes indicated a clear separation of samples based on their enrichment in CD8 CICs; CICs-low samples were enriched in memory and resting related genes versus cytotoxic and effector genes for CICs-high samples. Moreover, clustering on this basis further separated CIC-high samples based on their similarity to 28z or BBz CARs while highlighting

underlying transcriptional differences. Notably, despite having a CD8+ phenotype close to that of 28z, FCRL6-CD3G and CD28-FCGR2A variants showed reduced expression levels of proinflammatory factors such as IFNG, CCL3, CCL4, and CSF2. In addition CD223-CD79A and FCRL6-FCGR2A presented higher expression of memory markers, including TCF7 and CD27 while still presenting high levels of effector genes, two potentially interesting hybrid phenotypes.

A different approach used to analyze individual CAR variants employs gene-set scoring, which allows us to systematically screen variants for specific T cell signatures and compares them to the clinically used CARs. By defining gene sets based on the expression of T cell marker genes as well as previously reported associations to phenotype[21–23], such as our own CAR-induced signature obtained from the most differentially expressed genes in the CICs, one can identify CAR variants predicted to have specific transcriptional states that can guide candidate selection (Fig. 3c, Supplementary Fig. 9 and Supplementary Data 1).

### Mapping of CAR T cell scRNAseq data onto a patient-derived TIL dataset

In order to evaluate our CAR variants in a more translational context, we compared our in vitro generated CAR transcriptional phenotypes to clinically relevant T cell phenotypes, such as TILs. The presence of CD8+ TILs in solid tumors is associated with a better prognosis[24,25]. The complexity of the tumor microenvironment, consisting not only of malignant cells but by a constantly evolving aggregate of tumor, immune and stromal cells, their secreted factors and extracellular matrix, cannot currently be reproduced adequately by in vitro and in vivo mouse models. T cells that manage to infiltrate, survive the solid tumor microenvironment and exert cytotoxicity (tumor-reactive TILs) likely present desired properties that drive a successful antitumor response. This makes the diverse transcriptomic landscape of TILs that are found to be associated with tumor regression a valuable and suitable reference to compare CAR T cell phenotypes.

As a reference, we used the recent work of Liu et al.[26], which consisted of scRNA-seq data of CD8+ TILs recovered from patients with non-small-cell lung cancer (NSCLC) subjected to ICB therapy (anti-PD-1 antibody treatment)[26]. In this study, ICB promoted the accumulation of reactivated and newly infiltrated T cells which presented precursor exhausted (Texp) and terminally exhausted (Tex) phenotypes and correlated with a beneficial antitumor response. Mapping of our CD8+ CAR T cells onto TILs showed a high degree of overlap of the two datasets and, consistent with our previous results, CAR T cells annotated as CICs segregated from negative control cells (TCR-negative and unstimulated 28z CAR T cells) as opposed to non-CICs cells which shared a similar distribution (Fig. 4a). Unsupervised cell clustering identified two clusters (C8 and C12) that were selectively enriched in CICs cells and also harbored a large number of TILs (Fig. 4b, c). To understand what clinically-related phenotypes can be induced by our CAR variants, we examined the underlying characteristics associated with C8 and C12 TILs. We found that CXCL13, described by Liu et al.[26], to be exclusively and highly expressed in Tex cells and Texp cells following ICB treatment, to be amongst the most differentially expressed genes for both clusters (Fig. 4d, e). In addition C8 and C12 TILs presented different expression levels of co-stimulatory receptors and genes associated with T cell activation and tissue cell retention (Fig. 4e). This suggests that C8 and C12 both represent distinct phenotypes with a role in tumor infiltration and a positive antitumor response in lung cancer following ICB.

We next examined the enrichment of our library in the tumor-reactive TIL associated clusters C8 and C12 (Fig. 4f). Several CARs, including 28z had high enrichment within them. Variants such as 4-1BB-FCGR2A, CD244-K1, and CD28-FCGR2A displayed even higher overall enrichment than 28z. In addition, different C8 to C12 ratios to 28z, found in variants like CD150-CD79B and CD30-CD79B, present phenotypes that are worthy of further study. Overall, our mapping approach reveals additional CAR variants of interest possessing transcriptional phenotypes similar to ICB-treatment responsive TILs and thus may thus hold potential for the treatment of solid tumors, in particular NSCLC.

### In-depth functional characterization of selected CAR variants

Guided by scRNA-seq data and transcriptional mapping to TIL data, we selected ten CAR variants and the clinical benchmarks 28z and BBz for characterization with functional assays. The variants were rationally selected on the basis of their notable transcriptional and signaling-associated phenotypes (Fig. 5a, Supplementary Fig. 10): variants A (FCRL6-CD3G) and B (CD28-FCGR2A) were enriched in cytotoxic and terminal effector CD8 CICs and display an overall CD8+ transcriptional phenotype similar to 28z, though they showed reduced expression of CRS-associated factors with variant A displaying a potentially less terminally differentiated phenotype; variant C (4-1BB-FCER1G) presented a CD8+ phenotype closer to BBz CAR; D (FCRL6-FCER2A) and E (FCRL1-CD79B) showed increased expression of memory associated markers while still presenting enrichment in key CICs and tumor-reactive TIL associated clusters; variants F (4-1BB-FCER2A), G (CD150-CD79B), H (CD30-CD79B) and I (CD244-K1) all presented a high enrichment in tumor-reactive TIL associated cluster 8 and a different range of enrichments in cluster 12; variant J (TIM1-CD79B) showed no enrichment in CICs or tumor-reactive TIL associated clusters and maintained a closer phenotype to the unstimulated controls. As an additional control, we also sorted TCR-negative T cells. Flow cytometry analysis of CAR surface expression showed significantly different CAR expression levels across variants. With the exception of CAR E, variants were expressed at lower levels compared to the standard 28z CAR (Supplementary Fig. 11). Since all CARs are expressed from the same genomic locus, this may be the result of combining different cytoplasmic architectures with the CD28 transmembrane domain.

In order to determine the extent to which CAR variants influence T cell differentiation programs, we performed multi-parameter flow cytometry on 18 surface markers, including that characteristic of memory, activated and exhausted T cell phenotypes (Supplementary Fig. 12). Following 4-day co-cultures with SKBR3 (HER2-positive) tumor cells, surface expression profiles revealed moderate differences across variants and TCR-negative control cells. CD62L and CD45RA markers were used to determine different T cell differentiation states (Fig. 5b). Whilst CD4+ cells maintained a central memory phenotype in a high percentage of the cells for all samples, the CD8+ cell compartment showed more notable differences. CARs E, I and J showed a more effector memory and terminally differentiated phenotype closer to that of 28z CAR. BBz CAR on the other hand had a larger central memory compartment more similar to CARs A, G and H. Interestingly CAR F had a more expanded Tscm, comparable to that of TCR-negative cells. Regarding activation, the expression of early (CD69), middle (CD25) and late (HLA-DR) T cell activation markers indicate comparable levels of activation across CAR variants with moderate expression of early and very high expression of middle and late marker genes (Supplementary Fig. 13a). The TCR-negative control group had as expected reduced expression of both CD69 and CD25 but high expression of HLA-DR, which is probably a result of initial CD3/CD28 bead activation. CD39 and CD137 were expressed at low levels and only moderate differences were observed in the CD8+ cells. In order to assess T cell exhaustion, we measured the number of exhaustion markers (CTLA4, LAG3, TIGIT, TIM3, and PD1) simultaneously expressed per cell (Fig. 5c and Supplementary Fig. 13b). This number was considerably higher in all CAR variants compared to TCR-negative cells reflecting CAR induced T cell activation, however, no meaningful differences were found across variants except for those in the expression of single markers (Supplementary Fig. 9b), such as LAG3. Longer stimulation times would be needed in order to assess differences in exhaustion.

Next, we measured the cytotoxic capacity of the selected CAR variants for tumor cell control and elimination through live-cell imaging. As target cell lines, we used SKBR3 and MCF-7, the latter of which is another breast cancer cell line expressing HER2, albeit at lower level[27]. To track the cytotoxic activity via cancer cell death, we

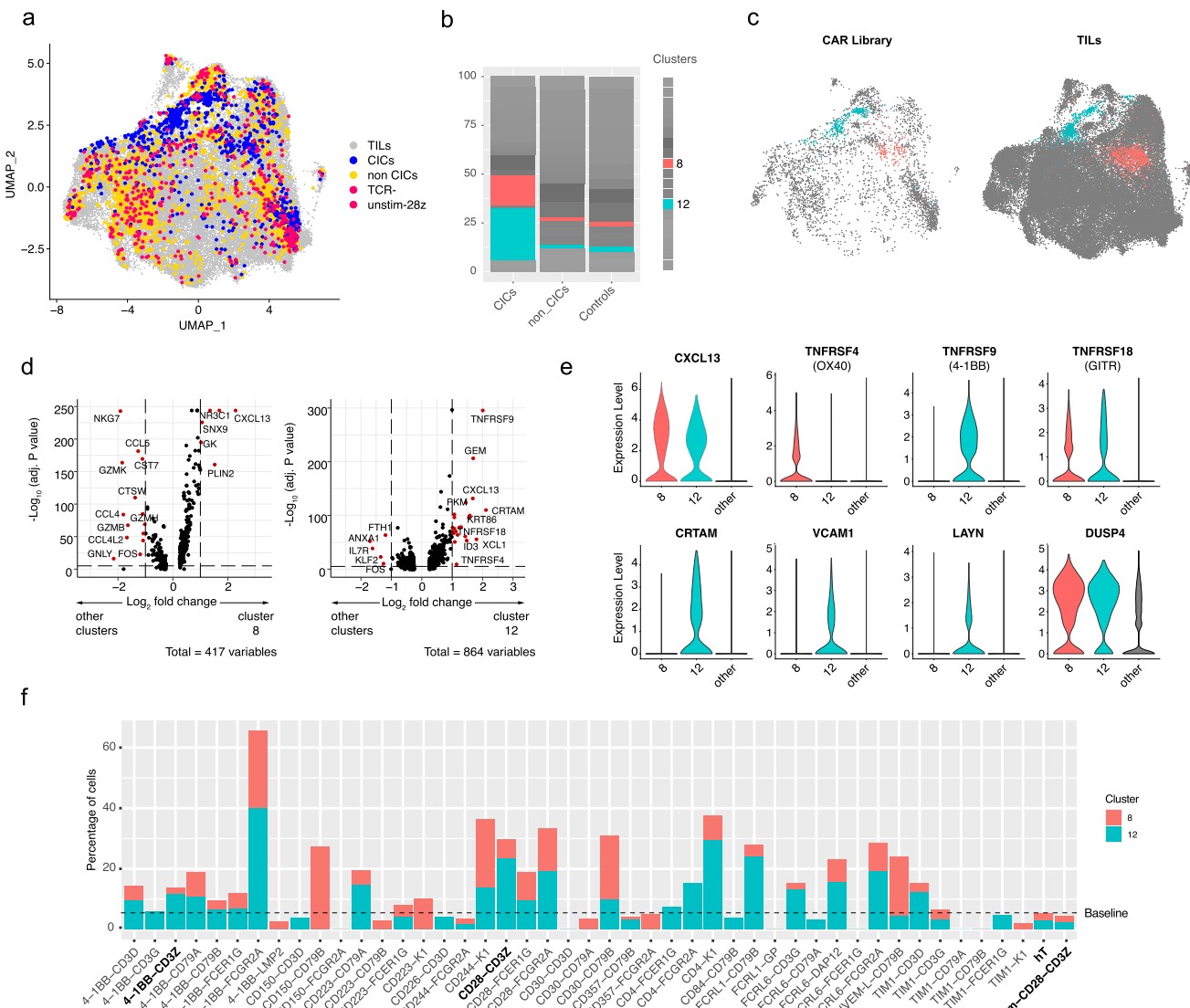

**Fig. 4 | Guiding the selection of CAR variants through transcriptional mapping on TILs. a** UMAP embedding resulting from the integration of CD8+ CAR library T cells with CD8+ TILs recovered from patients with non-small-cell lung cancer pre- and post-anti-PD-1 ICB treatment[26]. CAR T cells are highlighted and colored based on the cluster annotations from Fig. 2. CICs = CAR-induced clusters. **b** Enrichment of CAR library T cells across the 19 different clusters generated from unsupervised cell clustering. **c** UMAP plots highlighting cells from clusters 8 and 12. CAR library T cells and TILs are separated in two individual plots. **d** Volcano plots showing

differentially expressed genes for TILs between clusters 8 or 12 and all the rest of clusters. Statistical significance was determined though the adjusted *p*-values generated using the *FindMarker()* function of Seurat package (two-sided Wilcoxon Rank Sum test). Source data are provided as a Source Data file. **e** Violin plots showing the expression levels of a selection of marker genes (only TILs). **f** Enrichment of cells in clusters 8 and 12 across different CAR variants. Only variants with at least 20 CD8 T cells are shown. In all panels, TCR- refers to T cells without a TCR.

engineered GFP-expressing target lines using CRISPR-Cas9 HDR (Supplementary Fig. 14a). After three rounds of iterative sorting, the resulting populations showed stable expression of GFP in over 90% of cells (Supplementary Fig. 14b–e). For SKBR3, we tracked the total fluorescence of standard (sparse) co-cultures with the selected CAR T cell variants at various ratios (Fig. 5d, f and Supplementary Fig. 15a). At the lowest ratio of CAR-T to tumor cells (1:1) only most potent CAR variants (28z, BBz and CARs A-D) were able to control tumor growth. Nevertheless, CAR variants E-J still showed evidence of tumor killing as tumor growth was delayed. At the intermediate and high ratios (3:1 and 8:1) all CARs were able to control or eliminate SKBR3 expansion, however different killing dynamics could be observed: CARs A and D showed the fastest killing, followed by CARs B, C, 28z, and BBz, all of which had similar killing rates, CARs E and F which had only a slight delay over the benchmarks, CARs G-I and finally CAR J which was considerably the slowest. As an alternative cytotoxicity model, we

sought to challenge the CAR variants against three-dimensional cancer microtissues (spheroid structures). We established conditions under which the GFP+ MCF-7 cell line forms spheroids (Supplementary Fig. 14f, g), which were then cultured with CAR T cell variants at various ratios (Fig. 5e, g and Supplementary Fig. 15b). Within this spheroid setting, we observed faster-killing dynamics probably due to the co-localization and high density of tumor and T cells in a reduced compartment. Despite this, we were able to observe consistent results with SKBR3 killing. Again, CAR A showed the most potent killing followed by CARs B, C, 28z, BBz, D and E. As before and across different E:T ratios, CAR J was the slowest.

Next, following 4-day co-cultures with SKBR3 target cells, we profiled the secretion of a panel of eight cytokines across variants. The cytokines assayed were pro-inflammatory and/or Th1-associated (GM-CSF, IFNγ, TNFα, and IL12p70), Th2-associated (IL-4, Il-5, and IL-13) and IL10, which despite being considered an anti-inflammatory

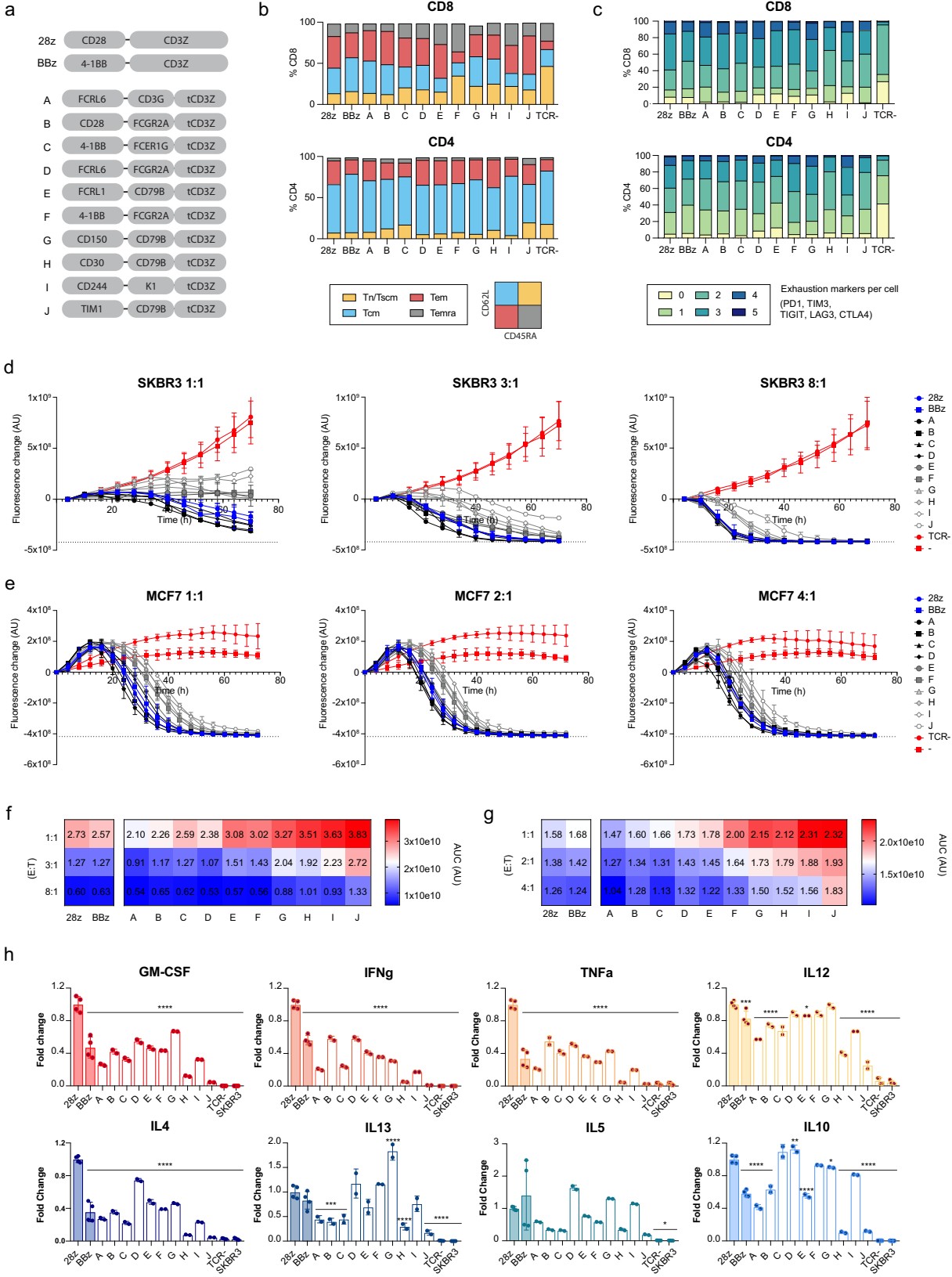

Th2-associated cytokine, it has a pleiotropic function in tumor biology and has been closely related to CRS in the context of CAR T cell therapy (Fig. 5h). As previously reported in literature, 28z CAR consistently secreted the highest levels of nearly every cytokine compared to BBz[28], this was particularly clear for the pro-inflammatory cytokines GM-CSF, IFNγ and TNFα. Notably, with few exceptions, this was also the case

when comparing 28z to the panel of ten candidate CAR variants. Amongst CARs A-D, which displayed similar or faster killing dynamics than the benchmark CARs, CARs B, and D showed strong similarity with BBz, occupying the middle range of cytokine secretion levels, whereas A and C showed even lower levels of secretion than BBz. CARs E, F, G, and I presented different patterns of intermediate secretion

**Fig. 5 | Functional characterization of selected CAR variants confirms their diverse phenotypes and potential enhanced properties. a** Ten CAR signaling domain variants were selected for individual characterization along with the clinically used 28z and BBz CARs. **b** Proportion of T cell differentiation subsets observed in CAR or TCR-negative (TCR-) T cells following a 4 day co-culture with SKBR3 cells (4:1 E:T ratio) as measured by CD62L and CD45RA surface expression (naive T cells (Tn), stem central memory T cells (Tscm), central memory T cells (Tcm), Effector Memory T cells (TEM), Effector Memory RA-positive T cells (TEMRA)). **c** Proportion of CAR or TCR- T cells simultaneously expressing different number of exhaustion markers (PD1, TIM3, TIGIT, LAG3, and CTLA4) following the co-culture conditions described in **b**. **d**, **e** CAR T cell-mediated cytotoxicity of HER2+/GFP+ tumor cells quantified over time by fluorescence microscopy. The fluorescence change values represent the difference in GFP intensity compared to time point 0, and a dashed line represents the baseline where no GFP+ cells are left. CAR T cells were co-

cultured at different E: T ratios with either SKBR3 adherent cells in a sparse 2D culture in **d** or with a single tumor spheroid of MCF-7 cells in **e**. **f**, **g** Heat maps comparing the area under the curve (AUC) of the killing curves in **d**, **e** across different CAR variants. **h** Cytokine secretion profile of a selection of 8 cytokines following a 4 day co-culture of CAR T cells and SKBR3 cells at a 4:1 E: T ratio. The levels of cytokines in the co-culture medium were quantified by fluorescence-encoded multiplex bead assays and are shown as fold change compared to the benchmark 28z CAR. To assess significant differences between each variant and 28z, a one-way ANOVA and Dunnett's multiple comparisons test was used with the following significance indicators: *$p$-value <0.05, **$p$-value <0.01, ***$p$-value <0.001 and ****$p$-value <0.0001. In all panels, TCR- refers to T cells without a TCR and error bars represent the S.D. ($n = 2$ independent technical replicates for variants A-J and $n = 2$ technical replicates from two independent experiments for control groups). Source and statistical data are provided as a Source Data file.

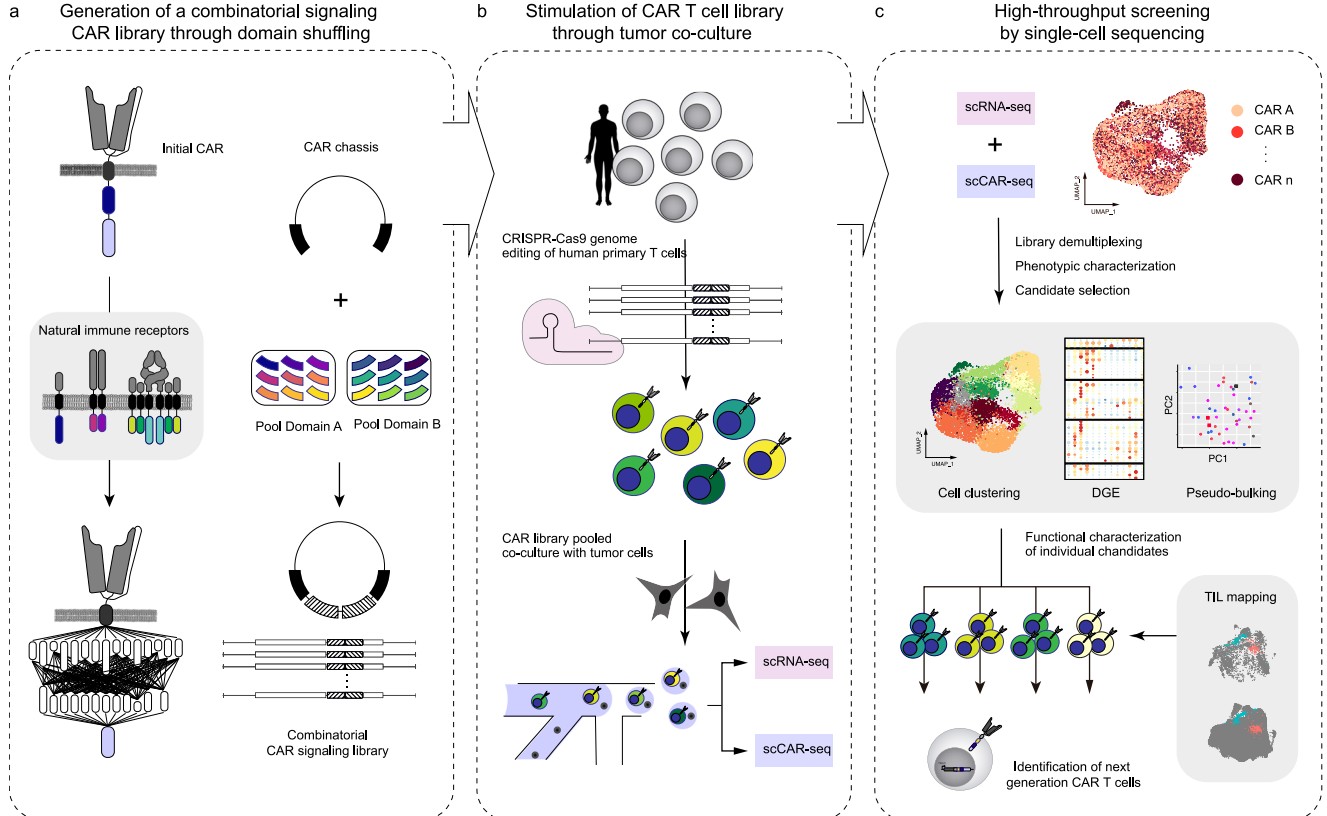

**Fig. 6 | Schematic overview of speedingCARs: an integrated approach for the rapid engineering of CARs. a** First, an initial CAR architecture is chosen based on functionality and encoded in a DNA plasmid. Next, one or more modular domains of the CAR are swapped for alternative natural immune intracellular signaling domain in a semi-random combinatorial fashion, resulting in a plasmid library of CAR variants with diverse combinations. **b** The plasmid library of CAR variants is genomically integrated at the *TRAC* locus of primary human T cells via CRISPR-Cas9

genome editing. This ensures that each cell expresses a single CAR variant, and simultaneously deletes the endogenous TCR. A pooled library of CAR T cells is co-cultured in the presence of tumor cells expressing cognate antigens. **c** The functional screening of the pooled CAR T cell library is performed by scRNA-seq and scCAR-seq, revealing the transcriptional phenotype. This dataset is used to select promising variants for in-depth characterization with functional assays.

levels characterized by high expression of IL-12 and IL-10 and significantly high levels of IL13 in variant G. CARs H and J consistently presented the lowest cytokine secretion resembling that of the unstimulated controls (TCR-negative and SKBR3 cells alone).

## Discussion

Here, we developed speedingCARs, an integrated method that combines signaling domain shuffling and single-cell sequencing to expand the range and functional profiles of CAR T cells (Fig. 6), which may eventually enable enhanced and personalized cell therapies. Our approach is designed for rapid and productive CAR engineering and utilizes two important concepts. First, CARs were conceived with modularity in mind, making them especially suitable for the

combinatorial shuffling of signaling domains derived from a wide range of receptor types. Second, scRNA-seq is currently feasible at an appropriate scale for screening a pooled library of CAR variants in order to identify transcriptional phenotypes with unique T cell activation profiles. To maximize translatability, we introduced the signaling domain-shuffled library into primary human T cells and triggered CAR activation through a co-culture with tumor cells expressing cognate antigen (HER2). This unique strategy allowed us to identify functional CARs with previously unused intracellular signaling domain combinations. These candidates exhibit properties that are uncommon in current standard designs and, thus may lead to new applications for CAR T cells. In addition, the depth and resolution of the output data allowed us to compare CAR T cell-induced phenotypes to

the transcriptional landscape found in clinically relevant settings, comprising the first steps towards customized CAR T cell therapies.

Among natural proteins, domain shuffling is thought to be a major evolutionary driver[29]. By definition, a domain's structure is modular and its function is portable. DNA translocation events that carry a domain-coding sequence into another gene can thus be well-tolerated from a functional standpoint. A domain might synergize with its neighboring domains, e.g. a kinase domain joining a binding domain, potentially creating a new pathway branch[30,31]. Signaling proteins are especially likely to be modular, leading to their embedding in complex networks of protein-protein interactions. With their signaling domains typically taken from T cell immune receptors, CARs are no exception. While the understanding of natural T cell signaling has benefited from decades of research, there is little knowledge on the effects of mixing and matching signaling domains in synthetic receptors such as CARs. Efforts to date have selected domain combinations largely by trial and error and from a small pool of known effectors[10,12,13]. Inspired by natural evolution, the speedingCARs method relies on random domain shuffling to generate a library of all possible combinations from which functional pairings can be identified.

While domain shuffling constitutes a powerful method for rapidly engineering new diversity in a protein, this can make the choice of screening strategy a challenging ordeal. Previous work for CAR engineering relied on functional screening based on the expression of single reporter genes or proteins (e.g., IL-2, NFAT, CD69) in immortalized cell lines[14,27,32–34]. While these approaches enable high-throughput screening of CAR libraries, they are limited by their unidimensionality: they generally reveal only a single aspect of the effector response and do not capture the full complexity of the deeply interconnected signaling network of T cell activation. Furthermore, immortalized cell lines do not fully recapitulate primary T cells, reducing the translatability of the screening results. Rather, to obtain multi-dimensional and translatable CAR functional profiles, primary T cells and RNA-seq offer a powerful alternative. The use of scRNA-seq, in particular is fast becoming an important tool in the characterization of CAR T cells before and during their clinical evaluation[9], and is especially suited for the screening of a pooled library of engineered cells. Recent work by Roth et al. demonstrated how scRNA-seq can be used to capture the transcriptome and the corresponding identity of a library member in engineered cells[35]. In their approach, a pooled library of 36 knock-in genes was screened in combination with a NY-ESO-1-specific TCR to find transcriptional phenotypes that could enhance the induced T cell response. As the capacity of scRNA-seq continues to grow, larger libraries can be screened in this way. Here, we harnessed current scRNA-seq capabilities to screen a library of 180 possible CAR signaling variants and directly identify unique transcriptional phenotypes of T cell activation. Our method circumvents the need for engineered cell lines and reporter genes, as it retains a robust throughput while greatly enhancing the translatability of the screening results.

Many of the pitfalls of CAR T cells are being addressed with complementary solutions such as additional gene editing to enhance the T cell immune response or the co-administration of immunomodulating compounds[4–8]. These solutions risk adding layers of complexity to what remains a challenging treatment to administer. Ideally, a single genome-integrated CAR would suffice, as in currently approved regimens, but different tumors may require slightly different approaches, making it challenging to find an optimal construct for a given situation. By shuffling signaling domains from diverse origins, we aimed to identify CARs that promote unique transcriptional phenotypes, as well as domains that may be associated with novel immunological profiles and functions. Indeed, we found significant phenotypic diversity in our CAR library, especially with genes related to effector and cytotoxic function, memory differentiation, Th1/Th2 classification and tumor infiltration. These unique profiles may prove

valuable in specific contexts where maximal cytotoxicity is not the only sought-after property. For instance, in clinical settings, BBz CARs have sometimes proven to be superior to the 28z combination with respect to T cell exhaustion and persistence in vivo[36,37], or showing a reduced incidence of adverse events[37]. Despite the limited translatability of in vitro assays to therapeutic efficiency and the lack of clinical validation of our tested candidates, the cytokine and cytotoxicity functional assays we performed suggest that all of our selected CAR variants have some ability to trigger a response compared to the controls. scRNAseq guided the rational selection of four CAR variants (A-D) that perform equally well or better than clinical benchmarks based on different in vitro killing assays and trigger distinct T cell activation phenotypes. Such phenotypes include in some cases the reduced secretion of proinflammatory molecules such as GM-CSF, IFNγ, and TNFα, where reduced expression has been positively correlated with safer and more effective CAR T cell products in different therapeutic settings[38–40]. Furthermore, we observed interesting patterns and synergies regarding the choice of CAR signaling domains; some signaling domains appear more productive than others. For instance, in terms of representation in the transcriptomics screen, the prominence of combinations featuring CD79B may reflect an ability to induce proliferation. In addition, Fc or Fc-like receptor domains were often present in variants with high enrichment in CICs. Following selection, we indeed found that variants A-F, all featuring these domains, induce the most potent tumor-killing dynamics. FcεRIγ, present in variant C, was one of the two original signaling domains used in early CARs, the other being CD3ζ[41]. Our results suggest that other Fc domains and combinations may be worth testing further. Likewise, CD30, a member of the Tumor Necrosis Factor Receptor (TNFR) superfamily and CD150, a member of the SLAM family may be responsible for the enhanced response of variants G and H over J. Little is known on the exact function of these receptors, however, the interaction of CD30 with TNFR-associated factors (TRAF)[42] or the signaling of CD150 though its immunoreceptor tyrosine-based switch motifs (ITSM)[43] may have a superior synergy with CD79B and CD3z signaling. In their flow cytometry-based CAR screening method, Goodman et al. also identified TNFR superfamily members as conferring functional enhancements[15], on the other hand SLAM receptor domains have not been studied in the context of a CAR. Lastly, we note that the presence of the CD28 and 4-1BB signaling domains among selected variants confirm their important co-stimulatory properties. However, other CAR constructs incorporating these domains (e.g. 4-1BB-LMP2 and 4-1BB-DAP12) showed poor T cell effector potential in the transcriptomics data and CAR F performed less effectively than BBz in functional assays, affirming that the contribution of all signaling domains in a CAR matters.

What constitutes the optimal T cell phenotype for adoptive cell therapies is an elusive question. The killing of tumor cells with in vitro assays and in vivo mouse models does not always translate to clinical efficacy in patients. This may be due to a limited understanding of other dimensions of the T cell response, such as differentiation-linked phenotypes and the interplay with the patient's immune system and tumor microenvironment. Furthermore, each clinical context might benefit from a therapeutic T cell product more precisely tailored for the given situation[44]. Due to the multi-dimensionality of scRNA-seq data, in the present study and as a proof-of-concept analysis, we attempted to bridge the gap between phenotypic characterization and clinically relevant T cell states by mapping our new CAR induced T cell phenotypes to the transcriptional landscape found in TILs isolated from lung cancer patients following successful ICB treatment. Similarity between CAR-induced phenotypes and TILs from responding patients uncovered CAR variants worthy of further development since they may have the potential to drive superior T cell responses for the treatment of solid tumors, a major challenge in the field. Indeed, our functional characterization showed that enrichment in tumor-reactive TIL associated clusters generally correlated with the selection of

functional CARs. As the scRNA-seq data analysis toolbox keeps evolving and the available datasets become more diverse in contexts[45], mapping of in vitro generated transcriptomes of CAR libraries to clinical reference atlases may guide the selection of variants with beneficial properties for precisely defined clinical indications.

Altogether the speedingCARs method offers a path towards the next step in personalized and precision medicine. This can be expanded to the now growing interest in using CARs against other indications, such as viral infections[46,47], autoimmune disease[48,49] or organ transplants[50]. In addition, the expression of CARs in other cells of the immune system, such as natural killer (NK) cells, macrophages or neutrophils, is also being considered[46,51–53]. The use of a CAR to direct a targeted antitumor response while also exploiting characteristics of these other cell types such as their natural tropism towards tumor sites, distinct cytokine secretion signature, antigen-independent tumor killing abilities as well as their lack of alloreactivity could help break through current CAR T cell therapy limitations[54].

## Methods

### Library cloning
The CAR signaling domain library was constructed in a DNA plasmid vector using a Type IIS restriction enzyme cloning strategy, as previously described[18]. Briefly, the vector was designed with a cloning cassette within a CAR chassis, as illustrated in Fig. 1c. In this chassis, the CD3ζ signaling domain was segmented, retaining amino acids 100 to 164 (amino acids 52 to 99 were used in the pool of domain B). The vector was digested with the Type IIS restriction enzyme AarI (Thermo Fisher) for 4 h at 37 °C and treated with Antarctic phosphatase (NEB) for 30 min at 37 °C. The signaling domains were amplified from synthetic DNA gene templates (Twist Bioscience) with primer pairs F1/R1 or F2/R2 (Supplementary Table 2) and digested with AarI. An equimolar mix of all domain fragments was prepared for ligation into the digested CAR chassis vector with T4 ligase (NEB). The ligated plasmids were transformed and amplified in chemically-competent E. coli DH5α cells (NEB).

### Primary human T cell isolation and culture
Buffy coats from healthy donors were obtained from the Blutspendezentrum (University of Basel) following the general consent guidelines approved by swissethics (Swiss Association of Research Ethics Committees). All recruited volunteers provided written informed consent. Peripheral blood mononuclear cells (PBMCs) were extracted from buffy coats using a Ficoll gradient. T cells were then isolated using the EasySep human T cell isolation kit (Stemcell) and activated with human T-activator CD3/CD28 Dynabeads (Gibco) at a bead:cell ratio of 1:1. Activated T cells were cultured in X-VIVO 15 (Lonza) supplemented with 5% fetal bovine serum (FBS), 50 μM β-mercaptoethanol, 100 μg/mL Normocin (Invivogen) and 200 U/mL IL-2 (Peprotech), thereafter referred to as T cell growth medium. After 2-3 days, the beads were magnetically removed.

### Primary human T cell genome editing
We adapted our previous CRISPR-Cas9 genome editing protocol[27] to introduce CAR genes at the TRAC genomic locus. Double-stranded HDR DNA repair template was produced through PCR amplification. The primers F3 and R3 (Supplementary Table 2) were used to amplify the CAR gene and homology arms whilst incorporating truncated Cas9 target sequences (tCTS) as described in[55]. The product was purified using a QIAquick PCR Purification Kit (Qiagen). The ribonucleoprotein (RNP) particles were assembled by first duplexing the CRISPR RNA (crRNA; Supplementary Table 1) and trans-activating CRISPR RNA (tracrRNA) (IDT) through co-incubation at a 1:1 ratio (95 °C for 5 min). After cooling, 4 μL of duplexed RNA (100 μM) were complexed with 2 μL of Cas9 protein (62 μM; IDT) at room temperature for 20 min.

To generate CAR library T cells, 2 μg of DNA repair template was added to 6 μL of RNP and diluted in 100 μL P3 nucleofection buffer (Lonza). This mixture was nucleofected with 2×10⁶ stimulated human primary T cells 72 h after bead stimulation, using the 4D-Nucleofector (Lonza) with the program EO-115. The cells were then immediately diluted in 600 μL of T cell growth medium. To generate individual CAR T cell variants, 0.4 μg of DNA repair template was added to 1.2 μL of RNP and diluted in 20 μL P3 nucleofection buffer (Lonza). This mixture was nucleofected with 1×10⁶ stimulated human primary T cells 48 h after bead stimulation, using the program EH-115. The cells were then immediately diluted in 150 μL of T cell growth medium.

### Cancer cell line culture and genome editing
HER2 expressing cell lines SKBR3 and MCF-7[27] were cultured in Dulbecco's Modified Eagle Medium (DMEM) (Gibco) supplemented with 10% FBS, 1% penicillin-streptomycin (Gibco) and 50 mg/mL Normocin (Invivogen), thereafter referred to as cell line growth medium. CRISPR-Cas9 genome editing in these cell lines was performed with RNP particles as described above with the following differences: the crRNA was specific for CCR5[56] (Supplementary Table 1); the nucleofection buffer was Dulbecco's phosphate-buffered saline (DPBS) (Gibco); the nucleofector protocol was EO-117 for SKBR3 and EN-130 for MCF-7; and the cells were diluted in cell line growth medium.

### Library sequencing
We used next-generation sequencing to characterize the diversity of the CAR signaling domain library. To examine the library at a plasmid level, we amplified the shuffled signaling domains using a PCR reaction with primers annealing to flanking sequences (F4 and R4; Supplementary Table 2) and purified the resulting product ranging between 304 and 1096 bp. To assess the library diversity following genome editing, we performed a two-step PCR. First, genomic DNA was extracted from 10⁴ to 10⁵ CAR-expressing T cells using QuickExtract buffer (Lucigen). The resulting product was used as a DNA template for a first PCR amplification reaction using F5 and R5 primers (Supplementary Table 2) to produce a long amplicon which confirmed TRAC locus genomic integration. This product was then used as DNA template for a second PCR amplification using primers F4 and R4. The final amplimers were purified and sequenced by long amplicon sequencing (PacBio) or Illumina paired-end sequencing (GENEWIZ).

### FACS of CAR expression and binding
Analytical and sorting flow cytometry protocols to confirm genomic integration of CAR constructs were described before and adapted here to primary human T cells[27] (Fig. 1f). The knock-out of the T cell receptor (TCR) was assessed with the absence of signal after staining 1:200 with CD3ε-APC (UCHT1,Biolegend). Each CAR construct contained a Strep tag which allowed for a two-step staining to validate successful knock-in; a 1:200 biotinylated anti-Strep tag antibody (GenScript) treatment was followed by a 1:400 Streptavidin-BrilliantViolet 421 conjugate (Biolegend). Similarly, HER2 binding was confirmed with 2.5 μg/mL soluble HER2 antigen (Merck) and subsequent 1:200 APC-labeled anti-HER2 antibody (Biolegend) incubation. Cells were washed in cold DPBS and kept on ice until analysis. CAR T cells were sorted into room temperature T cell growth medium and maintained for 5 days to recover before co-culture assays.

### Single cell sequencing and data analysis
Previously sorted library CAR T cells (rested for 8–10 days following bead removal) were co-cultured for 36 h with the high HER2-expressing tumor cell line SKBR3. An E:T ratio of 1:2 was used to maximize the contact of CAR T cells with their target antigen. Immediately after co-culture, CAR-T cells were sorted by FACS (Supplementary Fig. 3) and single-cell RNA sequencing was performed using the 10X Genomics Chromium system (Chromium Single Cell 3′

Reagent Kit, v3 chemistry; PN-1000075) following manufacturer's instructions. In short 4000–20,000 cells were resuspended in PBS and loaded into a chromium microfluidics chip. Following GEM formation, reverse transcription and cDNA amplification, 25% of the sample was used for 3' gene expression library preparation, including the incorporation of Chromium i7 multiplex indices (PN-120262). The resulting transcriptome libraries were sequenced using the Illumina NovaSeq platform. scRNA-seq data were generated from three individual donors.

### Single cell CAR sequencing (scCAR-seq)

In order to de-multiplex the CAR T cell library within the scRNA-seq data, a scCAR-seq strategy was developed (adapted from[35]; Supplementary Fig. 2). Using 40 μg of 10X cDNA, the cytoplasmic region of the 10X pooled barcoded CAR cDNA molecules was amplified using KAPA-HIFI, a Read1-p5 primer (F6) and a customized Strep-Tag specific Read2 primer (R6, Supplementary Table 2). Following a 10 cycle PCR (95 C for 3', [98 C for 20", 67 C for 30", 72 C for 60"] x10, 72 C for 2') and a X0.65 SPRI bead DNA clean up (AMPure XP, Beckman Coulter) a second PCR using a p5 primer (F7, Supplementary Table 2) and a i7-Read2 primer (Chromium i7 Multiplex Kit, 10X Genomics, PN-120262) were used to further amplify the genetic material for 15 cycles (95 C for 3', [98 C for 20", 54 C for 30", 72 C for 60"] x15, 72 C for 2'). CAR amplicons were then sequenced using PacBio SMRT sequencing platform and Biostrings R package was used to assign a CAR variant to each 10X single-cell barcode.

### Analysis of scRNA-seq data

The raw scRNA-seq data was aligned to the human genome (GRCh38) using Cell Ranger (10x Genomics, version 3.1.0) and downstream analysis was carried out using the Seurat R package (version 4.0.1[57]). Low quality cells were removed based on the detection of low and high number of UMIs (500 <nFeature_RNA < 10,000) and high percentage of mitochondrial genes (Percentage_MT < 15% of total reads). scCAR-seq results were then used to assign CAR variants to each sequenced cell, obtaining an assignment rate of 59% of cells, and only cells assigned to a single CAR variant (55%) were used for downstream analysis.

After QC and CAR T cell assignment a total number of 9193 stimulated library CAR T cells, 1093 stimulated 28z CAR T cells, 406 stimulated BBz CAR T cells, 3755 unstimulated 28z CAR T cells and 8828 stimulated TCR-negative T cells were obtained. Each dataset was log normalized with a scale factor of 10 000 and sample integration was performed applying the reciprocal PCA seurat pipeline using 2000 variable integration features. Integrated data were then scaled regressing out cell cycle genes and dimensionality reduction was done using the *RunPCA* function. *FindNeighbors* and *FindClusters* functions were used to do unsupervised cell clustering and differential gene expression (*FindAllMarkers*) was used to find marker genes used for cluster annotation. The results were then visualized using UMAP dimensionality reduction. Gene set enrichment analysis was carried out using gProfiler2 R package[58]. Pseudo-bulk samples were generated using the *AverageExpression* Seurat function on cells grouped by CAR variant metadata annotation. The normalized counts data was used to do PCA analysis (PCAtools) and the scaled data was used to look at the expression of marker genes. Gene set scores were computed using Ucell R package[59].

### Integration and analysis of TIL and CAR scRNA-seq data

The count scRNA-seq data from Liu et al., accessible under GSE179994 (GSE179994_all.Tcell.rawCounts.rds.gz) was downloaded and the following samples were selected s for downstream analysis: P1.tr.1, P1.tr.2, P1.tr.3, P1.ut, P13.tr.1, P13.tr.2, P13.ut, P19.tr, P19.ut, P30.tr, P30.ut, P36.tr.1, P38.tr.1. Data analysis and QC was performed as previously described using the Seurat R package (version 4.1.0[57]). In addition,

CD8+ cells were selected based on the lack of CD4 expression. Next, CAR and TIL seurat objects were joined and split by patient identity using the *SplitObject* function. To remove bias from cell cycle stages, a cell cycle score was assigned to every cell using *CellCycleScoring*. Before integration, *SCTransform* was used on each list object while regressing out Percentage_MT and cell cycle scores. Following the Seurat integration pipeline, 3000 integration features were selected using *SelectIntegrationFeatures*, and all seurat objects (CAR and TIL datasets) were combined using the *merge* function (using the previously selected 3000 integration features as variable features). Consecutively, *RunPCA* and *RunHarmony* from the Harmony R package (version 1.0[60]) were used to remove batch effects. Finally, we used *RunUMAP*, *FindNeighbors*, and *FindClusters* to generate a UMAP visualization and to perform unsupervised clustering of the integrated data.

### Cytokine secretion

The cytokine secretion of CAR T cells was measured following co-culture with SKBR3 target cells. For each replicate, 40,000 CAR T cells and 10,000 target cells were incubated in a volume of 200 μL of T cell growth medium supplemented with 50 U/mL IL-2 for 4 days. The culture supernatant was obtained by centrifugation and cytokine levels was analyzed using a Bio-Plex Pro Human Cytokine Th1/Th2 Assay (Bio-rad) according to manufacturer's instructions. Briefly, supernatants were co-incubated with washed magnetic fluorescent beads coated with capture antibodies for the analytes GM-CSF, IFN-γ, TNF-α, IL-2, IL-4, IL-5, IL-10, IL-12 (p70), and IL-13. Following washes, beads were co-incubated with PE-labeled antibodies specific to the analytes. After final washes, beads were analyzed using a Bio-Plex MAGPIX (Bio-rad). IL-2 data were excluded from the analysis due to the supplementation of this cytokine in the medium.

### Formation of MCF-7/GFP spheroids

Individual MCF-7/GFP spheroids were formed in ultra-low adherent, Nunclon™ Sphera™ U-shaped-bottom, 96-well plate (Thermo Fisher Scientific). In brief, cells were detached from the cell culture flask using 1X TrypLE™ Express enzyme (Gibco) and re-suspended at a density of $10^4$ cells/mL in pre-warmed complete minimum essential media (MEM) that contains 5 mg/mL human recombinant insulin (Gibco), 1X MEM non-essential amino acids (NEAA) (Gibco), 1 mM sodium pyruvate (Merck), and 50 mg/mL Kanamycin (BioConcept). 100 mL of cell suspension was loaded into each U-shaped-bottom well, resulting in an initial seeding of 1000 cells/spheroid. Cells were spun down at 250 g for 2 min and then kept in a humidified incubator at 37 °C and 5% $CO_2$ (Binder GmbH) without medium exchange for 3 days. Before each experiment, MCF-7 spheroids were imaged with a Cell3iMager Neo plate scanning system (SCREEN Group) for quality check. Compact MCF-7/GFP spheroids with a diameter of approximately 350 mm were qualified for further experiment.

### Live imaging of cytotoxicity

CAR T cells and GFP-expressing cancer cells were co-cultured in an incubated chamber equipped with a wide-field Nikon Ti2 microscope to visualize target cell death. For 2D SKBR3 killing, cells were mixed at designated ratios in a glass bottom 96-well plate in phenol red-free MEM supplemented with 1X NEAA (Gibco), 1X Glutamax (Gibco), 1 mM sodium pyruvate (Merck), 10% FBS (Gibco) and 50 U/mL IL-2 (Peprotech). For 3D MCF-7 killing, microtissues of $10^3$ cells were co-cultured at designated ratios in MT medium supplemented with 50 U/mL IL-2. Images were captured every 4 or 6 h for 72 h. For image analysis, a pipeline combining Ilastik (only for 2D SKBR3 killing) and Fiji was used to do background subtraction, segmentation and finally extract the total fluorescence of detected cell objects. The resulting values were analyzed with R and plotted with Graphpad (Prism).

## Multi-parameter flow cytometry for immunological markers

Following 4-day co-cultures of CAR T cells and SKBR3 cells at a ratio of 4:1, cells were extracted by centrifugation and prepared for flow cytometry analysis of immunological markers. First, cells were stained for viability (Zombie Aqua, Biolegend) in PBS, washed and stained in FACS buffer (2% FBS, 0.1% NaN3 in PBS) for the following markers: HLA-DR-Alexa Fluor 647 (L243), CD69-Pacific Blue (FN50), CD25-PE/Cy7 (M-A251), CD137/4-1BB-PE/Dazzle 594 (4B4-1), CD45RA- PE/Dazzle 594 (HI100), CCR7-APC/Cy7 (3D12), CD27-BV570 (O323), CD39-FITC (A1), CD127-PE (A019D5), CTLA-4-BV785 (L3D10), LAG-3-BV711 (11C3C65), TIGIT-BV421 (A15153G) from Biolegend; and CD3-BUV395 (UCHT1), CD4-BUV496 (SK3), CD8-BUV805 (SK1), CD62L-BV650 (SK11), PD-1-BB700 (EH12.1), TIM3-BV480 (7D3) from BD Biosciences (Supplementary Table 3). After washing, cells were analyzed using the Cytek Aurora full-spectrum flow cytometry technology. Data were further processed with FlowJo 10 software (BD Biosciences; Supplementary Fig. 12).

## Statistics & reproducibility

Statistical analysis was performed using Prism 9 software (GradPad) with the exception of sc-RNAseq data which was analyzed with R Studio using the packages mentioned above. No statistical method was used to predetermine sample size. When required, outliers resulting from technical problems were excluded from the analysis. Blinding was not relevant as data was quantified by software and not subject to investigators input. Except for scRNA-seq data, where randomization was used to obtain a balanced representation of CAR variants, experiments were not randomized. When micrographs are shown, at least three independent experiments were run, presenting similar results.

## Reporting summary

Further information on research design is available in the Nature Research Reporting Summary linked to this article.

# Data availability

All data are included in the Supplemental Information or available from the authors upon reasonable requests, as are unique reagents used in this Article. The raw numbers for charts and graphs are available in the Source Data file whenever possible. The raw and processed sequencing data generated in this study have been deposited in the Gene Expression Omnibus under accession number GSE214231 (https://www.ncbi.nlm.nih.gov/geo/query/acc.cgi?acc=GSE214231). Source data are provided with this paper.

# Code availability

The custom R scripts used for data analysis are publicly available at the GitHub repository: https://github.com/LSSI-ETH/SpeedingCARs_2022.

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

## Acknowledgements

We thank the ETH Zurich D-BSSE Single Cell Unit, the ETH Zurich D-BSSE Genomics facility and the ETH Zurich Functional Genomics Center Zurich for excellent support and assistance throughout this study. This work was supported by a ETH Zurich Post-doctoral Fellowship, Switzerland (to R.B.D.R.), Helmut Horten Stiftung, Switzerland (to S.T.R.) and NCCR Molecular Systems Engineering, Switzerland (to S.T.R.).

## Author contributions

R.C.R., R.B.D.R., and S.T.R. designed the study; R.C.R., R.B.D.R., F.S.S., and D.P. performed experiments; R.C.R. and F.B. performed and interpreted bioinformatic analyses; O.T.P.N. and E.K. assisted with obtaining biological material. R.C.R., R.B.D.R., F.B., F.S.S, D.P., O.T.P.N., E.K., H.L., A.H., N.K., and S.T.R. discussed results. R.C.R. and R.B.D.R. wrote the manuscript with input and commentaries from all authors.

## Competing interests

The authors declare no competing interests.

## Additional information

**Correspondence and requests** for materials should be addressed to Sai T. Reddy.

