## [Peer Review File · Nature Communications]

speedingCARs: accelerating the engineering of CAR T cells by signaling domain shuffling and single-cell sequencingEditorial Note: This manuscript has been previously reviewed at another journal that is not operating a transparent peer review scheme. This document only contains reviewer comments and rebuttal letters for versions considered at *Nature Communications*.

REVIEWERS' COMMENTS

Reviewer #2 (Remarks to the Author):

Overall, I think the revision has strengthened the manuscript and is suitable for publication. However, I would caution against making a strong claim about in vitro activity because the field as a whole has yet to establish any correlation between clinical efficacy and in vitro activity. For instance, I am uncertain whether the speed of in vitro cytotoxicity is a meaningful metric of clinical potency.

Reviewer #3 (Remarks to the Author):

The authors have significantly modified the text and added several important clarifying experiments. These new data have greatly improved the context and relevance of these findings. All of my concerns have been sufficiently addressed.

REVIEWERS' COMMENTS

Reviewer #2 (Remarks to the Author):

Overall, I think the revision has strengthened the manuscript and is suitable for publication. However, I would caution against making a strong claim about *in vitro* activity because the field as a whole has yet to establish any correlation between clinical efficacy and *in vitro* activity. For instance, I am uncertain whether the speed of *in vitro* cytotoxicity is a meaningful metric of clinical potency.

We appreciate the reviewer's comments, we directly address this point in the Discussion of our manuscript (Paragraph 5 lines 547-550). We agree that translatability of results from *in vitro* assays to clinical efficiency is still questionable (as this is also true to a certain extent for *in vivo* mouse models) and should therefore be interpreted with caution. Since our study did not perform clinical validation and in order to moderate the strength of our claims, we have added a remark in paragraph 4 of the Discussion (lines 519-421) to highlight the need to further validate the presented results in a more relevant clinical context before drawing solid conclusions.

Reviewer #3 (Remarks to the Author):

The authors have significantly modified the text and added several important clarifying experiments. These new data have greatly improved the context and relevance of these findings. All of my concerns have been sufficiently addressed.

We appreciate the reviewer's comments and are grateful of their support for publication.